# Feasibility of Using Seaweed *(Gracilaria coronopifolia*) Synbiotic as a Bioactive Material for Intestinal Health

**DOI:** 10.3390/foods8120623

**Published:** 2019-11-27

**Authors:** Po-Hsien Li, Wen-Chien Lu, Yung-Jia Chan, Yu-Ping Zhao, Xiao-Bao Nie, Chang-Xing Jiang, Yu-Xiang Ji

**Affiliations:** 1Department of Medicinal Botanical and Health Applications, Da-Yeh University, No.168, University Rd., Dacun, Changhua 51591, Taiwan; chanyungjia@gmail.com; 2College of Life Science and Food Engineering, Huaiyin Institute of Technology, Huaian 223003, Chinaniexiaobao1029@gmail.com (X.-B.N.); j1c2x3@gmail.com (C.-X.J.);; 3Department of Food and Beverage Management, Chung-Jen Junior College of Nursing, Health Sciences and Management, No.217, Hung-Mao-Pi, Chia-Yi City 60077, Taiwan; m104046@cjc.edu.tw

**Keywords:** *Gracilaria coronopifolia*, prebiotics, probiotics, synbiotics, gastrointestinal functions, seaweed

## Abstract

The market contains only limited health care products that combine prebiotics and probiotics. In this study, we developed a seaweed-based *Gracilaria coronopifolia* synbiotic and verified the efficacy by small intestinal cells (Caco-2). We also developed a functional material that promotes intestinal health and prevents intestinal inflammation. *G. coronopifolia* was used as a red seaweed prebiotic, and *Bifidobacterium bifidums*, *B. longum* subsp. *infantis*, *B. longum* subsp. *longum*, *Lactobacillus acidophilus*, and *L. delbrueckii* subsp. *bulgaricus* were mixed for the seaweed’s synbiotics. *G. coronopifolia* synbiotics were nontoxic to Caco-2 cells, and the survival rate was 101% to 117% for a multiplicative effect on cell survival. After cells were induced by H_2_O_2_, the levels of reactive oxygen species (ROS) increased to 151.5%, but after *G. coronopifolia* synbiotic treatment, decreased to a range between 101.8% and 109.6%. After cells were induced by tumor necrosis factor α, the ROS levels increased to 124.5%, but decreased to 57.7% with *G. coronopifolia symbiotic* treatment. *G. coronopifolia* synbiotics could effectively inhibit the production of ROS intestinal cells under oxidative stress (induced by H_2_O_2_ and tumor necrosis factor α (TNF-α)), which can reduce the damage of cells under oxidative stress. Functioning of intestinal cells could be improved by inhibiting the production of inflammatory factor substances (interleukin 8) with *G. coronopifolia* symbiotic treatment. Also, gastrointestinal diseases may be retarded by a synbiotic developed from *G. coronopifolia* to promote intestinal health and prevent intestinal inflammation.

## 1. Introduction

Seaweed has played an important role in traditional remedies for many centuries for treating hyperthyroidism and other glandular diseases. Recent studies show that algae have compounds with cytostatic, antiviral, antifungal and antibacterial activities [1,2,3]. *Gracilaria coronopifolia* (*G. coronopifolia*), belonging to the family Gracilariaceae, is one of the most common types of eatable endemic Hawaiian marine red seaweed [4]. *G. coronopifolia* is often red but may turn to pink or reddish white under bright sunlight. The species has salt tolerance ability due to the cytoplasm, which will adjust the osmolarity automatically to accomplish the osmolarity of the sea environment. *G. coronopifolia* and other species of *Gracilaria* are also called by the Japanese name “ogo” as a food resource that is rich in minerals, polysaccharides, and vitamins but low in calories. The natural products and metabolites isolated from marine seaweed are important sources of bioactive compounds that could be developed into treatments for some diseases.

The World Health Organization (WHO) indicated that between 2008 and 2030, the number of global cancer deaths will increase by 45%. Most cancer deaths are due to lung, breast, colorectal, stomach, and liver cancer. The WHO, via the International Agency for Research on Cancer (IARC), maintains a classification of cancer-causing agents such as aging and environmental factors including physical carcinogens (ultraviolet and ionizing radiation), chemical carcinogens (tobacco smoke, food and water contaminants), and biological carcinogens (infections by certain viruses, bacteria, or parasites). Foods that combine nutrients and healthful substances for body maintenance needs and also the quality of foods are vital factors that affect the healthiness and composition of the gastrointestinal tract, the microbes present in the gastrointestinal tract, which are closely related to the health of the human body. Foodstuffs consumed will affect the distribution of intestinal flora, metabolic activity, and gastrointestinal environment.

The internal environment of the gastrointestinal tract is the main habitat of microorganisms in the human body. The health effects of microorganisms in the intestine can be divided into nutritional functions, resistance to infection by foreign pathogens, and reduction of contact with toxic substances [5]. For example, probiotic fermented milk containing *Lactobacillus* and *Bifidobacterium*, was reported to have a body health-promoting effect linked to enhancing the intestinal tract environment, including ameliorating inflammatory bowel disease and strengthening the immune system [6,7,8].

Probiotics are defined as viable microorganisms that, with an adequate amount, have a beneficial effect on intestinal health [9]. Probiotics, especially *Lactobacili* and *Bifidobacteria*, were reported to play an important role such as alleviating lactose intolerance [10], treatment of diarrhea [11], as well as immunomodulation [12], antimutagenic [13] and anticarcinogenic effects [14]. Prebiotics grouped into a category as a nutritional functioning compound are substances that can be used by probiotics (specific beneficial gut bacteria) to promote the balance of the flora in the intestine, such as oligosaccharides and dietary fibre, hence enhancing the health of the host [15]. Synbiotics are a combination of probiotics and prebiotics that can exert the physiological activity of probiotics, selectively increase the amount and survival rate of probiotics, prolong probiotic activity in the gastrointestinal gut, and increase the adaptability of probiotics, due to the complementation and intercorrelation of each other [16,17].

Previous studies have shown that when the body is exposed to extensive oxidative stress, it generates free radicals and reactive oxygen species (ROS), which have a great impact on human health [18]. Excessive ROS production may damage tissue, resulting in an inflammatory reaction that slowly leads to aging and some chronic diseases. High ROS levels in intestinal cells may trigger the pathogenesis and progression of incurable chronic inflammatory bowel disease (IBD) [19,20]. Previous studies found that oxidative stress signaling is involved in and contributes to the development of IBD at multiple levels of function [21]. Therefore, the health of the intestinal tract can be maintained by regulating the production of ROS in the intestinal tract, thus preventing the inflammatory reaction caused by oxidative stress. The small intestinal cells (Caco-2) line derived from colon carcinoma cells is commonly used as a model of the gastrointestinal epithelial barrier to evaluate the function of the human gastrointestinal tract [22] and to investigate the potential toxic effects of functional foods with new formulations [23].

Earlier studies indicated that seaweed is full of dietary fibre, protein, minerals (potassium, calcium, magnesium, and manganese) and omega-3 polyunsaturated fatty acids; meanwhile, red seaweed may contain high amounts of panthothetic acid, folic acid, and folinic acid as compared with brown seaweed [24]. Hence, *G. coronopifolia* could be a source of marine-based prebiotics to combine with probiotics for developing *G. coronopifolia* synbiotics. Such synbiotics may have an effect towards gastrointestinal diseases caused by improper diet or human aging and also promote intestinal health. However, because most gastrointestinal health products are usually separated into single prebiotics or probiotics and the combination of the two products (synbiotics) is relatively rare, here we developed marine-based *G. coronopifolia* synbiotics and by used the intestinal Caco-2 cell line model to verify their efficacy for developing functional materials that promote intestinal health and prevent intestinal inflammation.

## 2. Materials and Methods

### 2.1. Sample Preparation

*G. coronopifolia*, as marine-based seaweed extract samples, were purchased from the Huaian seafood market and pre-treated with washing, freeze-drying and pulverization. *G. coronopifolia* dried powder was added with distilled water in a 50:50 ratio and was heated to 60 °C for 6 h, then centrifuged at 8000 rpm for 15 min to separate the upper layer, then dried by a vacuum freeze-drier. The dried extract was collected and stored in a freezer in double-bagged polyethylene [25].

### 2.2. Bacterial Strains and Culture Conditions

Five probiotic strains, including *Bifidobacterium bifidums* (BCRC 11844), *Bifidobacterium longum* subsp. *infantis* (BCRC 14602), *Bifidobacterium longum* subsp. *longum* (BCRC 11847), and *Lactobacillus acidophilus* (BCRC 10695), *Lactobacillus delbrueckii* subsp. *bulgaricus* (BCRC 16053), were purchased from the Bioresource Collection and Research Centre of the Hsinchu Food Industry Research and Development Institute. *Bifidobacterium bifidums*, *B. longum* subsp. infantis, *B. longum* subsp. *longum*, *Lactobacillus acidophilus*, and *L. delbrueckii* subsp. *Bulgaricus* with the ratio of 20:20:20:20:20, were cultured on sterile reinforced clostridial medium (RCM), Man Rogosa Sharpe medium (MRS), and MRS + 0.05% cysteine medium at 37 °C until the absorbance wavelength at 610 nm reached 0.8 (1.0 × 10^8^ CFU/mL), suitable for further analysis [26].

### 2.3. G. coronopifolia Synbiotic Preparations

*G. coronopifolia* synbiotics (GS) were prepared by thoroughly mixing lyophilized powder of *G. coronopifolia* extract and five bacterial strains in different proportions, as GS1 (Strains: dried extract = 30:70); GS2 (Strains: dried extract = 50:50); GS3 (Strains: dried extract (70:30). While the five probiotic strains were with the ratio of 20: 20: 20: 20: 20, corresponding to 1.0 × 10^8^ CFU/mL [26].

### 2.4. Cell Culture

Two cell lines were used in this experiment: human intestinal (human colon adenocarcinoma, a clone of Caco-2) and human monocyte (human acute monocytic leukemia, THP-1). Caco-2 cells were cultured in Dulbecco’s Modified Eagle Medium (DMEM) (glutamine, high glucose, Sigma) containing 10% fetal bovine serum (FBS) and supplemented with 1% of antimicrobial (Antibiotic-Antimycotic, 100×, 15240062, Gibco USA) at 37 °C with 5% CO_2_. Caco-2 cells were propagated and maintained under standard cell culture conditions until the monolayer formed. Confluent Caco-2 cells were trypsinized by absorbing the supernatant, and cells were washed three times with 10 mL phosphate-buffer saline (PBS), followed by evenly dispersing 1 mL trypsin in the cell layer. After 5 min incubation, 1 mL FBS was added, and cells were aspirated and washed several times with 10 mL PBS to separate cells. The DMEM of Caco-2 cells were changed every 2 to 3 days.

THP-1 cells were cultured and maintained in RPMI medium containing 10% FBS at 37 °C, with 5% CO_2_. The medium was changed after centrifugation at 1000 rpm for 5 min; after removing the supernatant, fresh medium was added and mixed, then transferred to a new flask. After culture for 2 to 3 days, the cells could be further be a sub-cultured.

### 2.5. Co-Culture of Caco-2 and THP-1 Cells

A co-culture system of Caco-2 and THP-1 cells was used with modifications (Figure 1) [27]. In brief, Caco-2 cells were maintained in DMEM at 37 °C under humidified 5% CO_2_ and seeded in the upper chamber of a Transwell filter (Corning CoStar Corp., Cambridge, MA, USA) for 3 weeks (48–60 passages) until transepithelial electrical resistance (TEER) was observed. The upper-chamber Caco-2 cells were then placed in the lower chamber of the Transwell filter preloaded with THP-1 cells and incubated for 24 h. The upper layer was Caco-2 cells that had been cultured to form a monolayer cell membrane, and the lower layer was THP-1 cells, which were stimulated with 10 ng of lipopolysaccharides (LPS) per mL and 10 ng tumor necrosis factor α (TNF-α) per mL for 24 h. The lower cell culture medium was collected for experiments. An amount of 500 ng/mL LPS was used to verify the feasibility of the Caco-2 and THP-1 co-culture system for 3 h. Fluorescein isothiocyanate- dextran tracer (FD-4) (4 kDa, MW4400) was used as the positive control for calculating the penetration rate, and the TEER value before and after the test was detected.

### 2.6. Cell Viability Assay (MTT Assay)

After cells were collected by centrifugation at 1500 rpm for 5 min, cell density was calculated by using a cell counter, and cells were adjusted to 2 × 10^5^ cells/mL according to the value. Cells were cultured at 100 μL/well in 96-well plates. After incubation at 37 °C with 5% CO_2_ for 24 h, the original culture solution was eliminated, and 100 μL of different concentrations (0.1–1000 μg/mL) of sample solutions was added, then continued to incubate at 37 °C with 5% CO_2_ for 24 h. Next, 100 μL MTT (3-4,5-dimethylthiazol-2-yl-2,5-diphenyl tetrazolium bromide) was added to each well, and the reaction was performed at 37 °C for 4 h, and finally, 100 μL DMSO was added. The cell membrane was lysed and uniformly mixed to ensure complete dissolution of the blue-violet crystal. After standing for 40 min, the absorbance wavelength 570 nm was read by an ELISA reader. The sample was replaced with the culture medium as the control group and set to 100%, and the other blank group was the culture medium containing no cells and samples [28]. We calculated cell viability as
Cell viability (%) = ((sample group’s absorbance × 100)/control group’s absorbance)(1)

### 2.7. Single-Layer Cell TEER Determination

TEER provides ion flow resistance information across a single monolayer cell that correlates with the integrity of tight junctions between cells [29]. Caco-2 cells were inoculated in Millicell cell culture insert plates (PCF membrane, 0.4 μm pore size), the growth of monolayer cell membrane was observed, and the TEER value was measured at different times with the Millipore ERS resistance meter, for 21 days. A pair of electrodes was placed on the top and bottom of the cell monolayer when measuring the TEER, and the actual resistance value was measured by using a resistance meter and calculated as follows:TEER = (R_total_ − R_blank_) × A(Ω × cm^2^)(2)
where R_total_ is the measured resistance value, R_blank_ is the blank film resistance value, and A is the membrane area.

### 2.8. Determining ROS Production

The experimental sample, hydrogen peroxide (H_2_O_2_) at a final concentration of 400 μM, and 5 × 10^5^ cells/mL were added to 12-well plates (containing 2% serum medium) and incubated in a constant temperature incubator at 37 °C, with 5% CO_2_ for 12 h. At 30 min before the end of the reaction, 20 μM of the fluorescent dye 2’,7’-dichlorofluorescin diacetate (DCFH-DA) was added into the culture medium. After the nonpolar substance DCFH-DA infiltrates into the cells, it can be transformed by intracellular esterification to the non-fluorescent polar substance DCFH; DCFH is oxidized by H_2_O_2_ in the cell after it enters the cell and becomes a fluorescent substance, 2’,7’-dichlorodihydrofluorescein (DCF), for indirect quantification of intracellular ROS production. After the cells and DCFH-DA were reacted at 37 °C for 30 min, they were centrifuged for 10 min, 10,000 rpm and the supernatant was removed, then cells were further washed twice with PBS, suspended in 0.5 mL PBS, and placed in a 96-well black plate with a fluorescence analyzer (Hitachi F-4500, Hitachi Co. Ltd., Tokyo, Japan) for detecting fluorescence response (excitation 485 nm, emission 527 nm) [30].

### 2.9. Interleukin-8 (IL-8) Analysis

The cultured Caco-2 cell concentration was adjusted to 3 × 10^5^ cells/mL; cells were cultured in a 6-well plate for 21 days, and co-cultured with THP-1 cells that differentiated by using phorbol-12-myristate-13-acetate, 200 nM for 4 days. An amount of 1.5 mL of different concentrations (0.1–1000 μg/mL) of samples was injected into the inner chamber, and 2 mL of RPMI medium was added to the outer chamber, and further incubated in a 37 °C, 5% CO_2_ cell incubator for 3 h. Next, 100 ng/mL of TNF-α was added into the outer chamber for 24 h. The suspension was collected and stored at −20 °C. The cytokine human IL-8 was detected by using a human assay kit (431505).

A 96-well test plate was prepared with a monoclonal antibody, and 100 μL capture antibody diluted with PBS was added to each well, placed overnight at a temperature of 4 °C. The solution in each well was aspirated, then rinsed with wash buffer. An amount of 200 μL blocking buffer was added into each well and reacted at room temperature for 1 h. The aspiration of the solution was continued in wells and rinsed with wash buffer, then the test plate preparation was completed. An amount of 100 μL medium was added to the test plate and reacted at room temperature for 2 h, the medium in the well was aspirated and washed with wash buffer. An amount of 100 μL detection antibody was added in and reacted for 1 h under room temperature.

After the solution in each well was aspirated and washed with wash buffer, 100 μL horseradish peroxidase-conjugated streptavidin was added and further reacted at room temperature for 30 min. The solution in each well was aspirated and washed with the wash buffer, then 100 μL substrate solution was added into the test plate. The test solution was reacted at room temperature for 20 min in the dark, and 100 μL of the stop solution was added for terminating the reaction. The absorbance of each well was measured by using an ELISA reader at a of wavelength 450 nm.

### 2.10. Statistical Analysis

Experimental tests and treatments were carried out in triplicate. Data was reported as a mean (SD). All data were analyzed by using single-factor of ANOVA, with the F-value significant at *p* < 0.05), then Duncan’s new multiple range test was used to compare treatment means.

## 3. Results and Discussions

### 3.1. G. coronopifolia Synbiotic Synthesis and Storage Test

We used extracts of *G. coronopifolia* as a red seaweed prebiotic, mixed with five probiotics of *Lactobacilli* and *Bifidobacteria*, and supplemented with 15% skim milk powder as a protective agent, with the probiotics and prebiotics blending ratio of GS1 (30:70), GS2 (50:50), and GS3 (70:30). The number of bacteria of probiotic products containing live bacteria needs to be ≥10^6^ cfu/g to exert their activity; thus 10^6^ cfu/g is used as a screening condition for storage [31]. The storage stability test of the prepared *G. coronopifolia* synbiotics (GS1, GS2, and GS3) was performed at different temperatures (4 °C and −20 °C). With storage temperature 4 °C for *G. coronopifolia* synbiotics for 60 days, the number of bacteria remained at 10^9^ cfu/g (Figure 2a); after storage at −20 °C for 60 days, the number of bacteria was still maintained at 10^6^–10^8^ cfu/g for GS3 and GS1 synbiotics (Figure 2b). Thus, low temperature effectively delayed the reduction in number of bacteria in the *G. coronopifolia* synbiotics but retained the stability of the cryopreservation. The intestinal chronic diseases, such as the inflammatory bowel disease, colon cancer, and contentious bowel syndrome, are highly related to the microbial community of the intestinal tract and balanced activities [32]. Thus, the product stability needs to be maintained to ensure the superiority and quality of the *G. coronopifolia* synbiotics.

### 3.2. Survival of G. coronopifolia Synbiotics in Caco-2 Cells

To explore *G. coronopifolia* synbiotics for preventing intestinal inflammation, we used an intestinal cell strain, the human intestinal cell line Caco-2. We examined cell viability after treatment with the combination of probiotic and the *G. coronopifolia* extract for 24 hs. At 0.1–1000 μg/mL, cell viability was 99% to −110%, 104% to −112% and 93% to −106%, respectively. At this concentration, the probiotic and *G. coronopifolia* extracts did not cause any harmful effects in intestinal cells. Caco-2 cells were used for toxicity tests of *G. coronopifolia* synbiotics. The survival of Caco-2 cells with 0.001 to 1 mg/mL of the GS1 synbiotic was 102% to −115%, GS2 synbiotic 98% to −106%, GS3 synbiotic 101% to −117%. Thus, the *G. coronopifolia* synbiotics were not toxic to Caco-2 cells.

### 3.3. Establishment of Caco-2 Cells and THP-1 Macrophage Co-Culture Mode

In this study, Caco-2 cells were used for evaluation, and the degree of change in trans-epithelial/endothelial electrical resistance (TEER) was measured. The value should be ≥600 Ω·cm^2^ to authorize the formation of a monolayer film. However, the TEER value for cultured Caco-2 cells until day 14 was 685 Ω·cm^2^ (Figure 3a); the required incubation time for forming a completed monolayer was observed by microscopy as day 21, 1053 Ω·cm^2^.

THP-1 cells are established from monocytes in the blood of leukemia patients. They secreted inflammatory cytokines and are appropriately induced to transform into macrophages. They are a good source to substitute cells for in vitro human mononuclear macrophages. We used cultured THP-1 human monocytes to establish the in vitro pattern of differentiation into macrophages. THP-1 cells were differentiated with 200 nM phorbol-12-myristate-13-acetate under microscopy observation. After 4 days, the cell morphology was transformed into a macrophage-like mode. The two cell types were co-cultured on a plug-in cell culture dish with the upper layer of Caco-2 cells and the lower layer of THP-1 macrophages for subsequent intestinal inflammatory evaluation.

### 3.4. Effect of G. coronopifolia Synbiotics on Intestinal Monolayer Membrane

Under the Caco-2 cells and THP-1 macrophages co-culture system, *G. coronopifolia* synbiotics were added to the upper layer of cells. At certain time intervals, the lower cell culture medium was sampled, and the permeability of the samples toward intestinal cells was observed. THP-1 cells were induced to secretes TNF-α, causing Caco-2 cell-surface damage after 24 h and TNF-α secretion was increased significantly. The production of TNF-α induced the Caco-2 cells to secrete interleukin-8 (IL-8) (an inflammatory factor that can induce an inflammatory reaction), which verified that inflammatory intestinal cells were induced by LPS and TNF-α and could promote the secretion of inflammatory factors with the co-culture system. Co-culture was feasible as an experimental model of intestinal inflammation. Co-cultured cells were treated with *G. coronopifolia* synbiotics at 0.001–0.1 mg/mL. The TEER value for 0.1 mg/mL GS1 synbiotic relative to FD-4 (100%) was 86% at the beginning of the test and increased to 95% after 180 min. Concurrently, TEER values at 0.001, 0.005 and 0.05 mg/mL were 105%, 103% and 101%, respectively, after 180 min (Figure 3b–d). Thus, the duration of action lasts for 3 h, whereas the TEER values for the *G. coronopifolia* synbiotics were still ≥ 80%. Therefore, the *G. coronopifolia* synbiotics did not damage the single layer film of intestinal cells [33]. Seaweed, especially red seaweed, contains a huge amount of dietary fibre, with an ability to lower cholesterol levels and act as a natural marine-based antioxidant in healthy food products [34]. The lipid absorption-restricted properties of seaweed endow them with the potential as a food additive for reducing calories, body weight control, and preventing of gastrointestinal diseases [35].

### 3.5. Protective Effects of G. coronopifolia Synbiotics on THP-1 and Caco-2 Cells under Oxidative Stress

The oxidative stress of THP-1 cells was induced by H_2_O_2_ and TNF-α at different concentrations (0.125–2 mM). Cell viability was verified after 24-h co-culture (Figure 4). Cell viability was negatively related to H_2_O_2_ concentration (Figure 4a). The cell viability of THP-1 cells gradually decreased from 95.5% (0.125 mM H_2_O_2_), 83.3% (0.25 mM H_2_O_2_) to 79.0% (0.5 mM H_2_O_2_). After 1 mM H_2_O_2_ treatment, the cell survival rate was 67.4%, which produced oxidative stress on cells but did not kill them. Figure 4b shows the MTT findings of the viability of THP-1 cells with 1–20 ng/mL TNF-α treatments. The survival rate was determined after 24 h of co-culture. The survival rate of THP-1 cells steadily decreased from 98.1% to 75.9% with TNF-α content 1 and 20 ng/mL, respectively, which also had a negative dose reaction. Therefore, 1 mM H_2_O_2_ and 20 ng/mL TNF-α were used to induce oxidative stress.

The culture system of *G. coronopifolia* synbiotics and THP-1 cells was used with of 1 mM H_2_O_2_ and 20 ng/mL TNF-α to induce oxidative stress formation, followed by cell viability tests. The viability of THP-1 cells after 1 mM H_2_O_2_ treatments was reduced to 70.7% (Figure 4c). Also, after the addition of *G. coronopifolia* synbiotics, the survival rate was increased. With 0.1 mg/mL GS1 synbiotic, the cell viability increased to 95.8% (0.005 mg/mL), slightly decreased to 85.2% (0.05 mg/mL) and increased to 100.3%. Likewise, the GS2 synbiotic also slightly changed THP-1 cell viability. GS2 synbiotic at 0.05 mg/mL conferred a higher cell viability rate (105.5%) and 103.9% at 0.1 mg/mL. However, 0.5 mg/mL of GS3 synbiotic increased cell viability to 81.2% (0.001 mg/mL), which steadily decreased to 70.7% (0.05 mg/mL) and increased to 75.4%. Concurrently, cell viability decreased to 74.4% under 20 ng/mL THF-α induction (Figure 4d). The cell viability peaked at 92.2% with 0.5 mg/mL GS1 synbiotic. However, 0.05 mg/mL GS2 synbiotic conferred the highest cell viability (94.2%). Nevertheless, 0.005 mg/mL GS3 synbiotic conferred 82.4% of THP-1 cell viability. *G. coronopifolia* synbiotics effectively reduced the cell damage caused by oxidative stress (H_2_O_2_ and TNF-α) and increased the cell survival rate. GS1, GS2, and GS3 at 0.1, 0.05, and 0.005 mg/mL were used for subsequent tests.

Figure 5 shows the cell viability of Caco-2 cells after induction with different doses of H_2_O_2_ (0.5–5 mM). The cell viability of Caco-2 cells gradually decreased from 94.1% (0.5 mM H_2_O_2_) to 84.9% 1 mM H_2_O_2_ and greatly decreased after induction with 2 mM H_2_O_2_ (48.7%). Hence, 0.5 mM H_2_O_2_ was used for the following ROS production test (Caco-2 cells) instead of 1 mM H_2_O_2_ because Caco-2 cells were suspended at the higher dose of H_2_O_2_. ROS production in Caco-2 cells with 0.5 mM H_2_O_2_/ 20 ng TNF-α, then treatment with *G. coronopifolia* synbiotic, is presented in Table 1. In the beginning, the amount of ROS produced was increased to 105.4% (100% for the control sample) after induction with H_2_O_2_. After *G. coronopifolia* synbiotic (GS1, GS2, and GS3) treatment, the amount of ROS produced was decreased greatly. At 0.05 mg/mL, ROS production decreased sharply to 63.3% and 48.8% with GS1 and GS2 synbiotics, respectively. In addition, 0.01 mg/mL GS3 synbiotic greatly decreased ROS production to 56.2%. Yet, after induction with 20 ng/mL of TNF-α, ROS production increased to 124.5% (100% for the control group). The amount of ROS produced decreased significantly with 0.005 mg/mL GS3 synbiotic (57.7%). Moreover, 0.01 mg/mL GS1 and 0.001 mg/mL GS2 also successfully decreased the ROS production to 93.0% and 60.6%, respectively. *G. coronopifolia* synbiotics capably slowed the production of ROS induced by H_2_O_2_ and TNF-α and reduced the oxidative stress of Caco-2 cells, which is expected to prevent the inflammatory reaction caused by oxidative stress. 4.0 mg/kg marine seaweed (*G. verrucosa*) successfully enhanced the immunity system of the rats and stimulated phagocytosis of the cells [36]. The total dietary fibre content of *G. coronopifolia, G. parvispora, G. salicornia* and *G. tikvahiae* was determined to be 27.0, 26.4, 35.8 and 28.4, respectively [37]. For example, 0.5 cups of fresh *Gracilaria* seaweed, is equivalent to 5–10 g total dietary fibre. Furthermore, rats fed a 0.5 mg/kg dose of water extract of *G. verrucose* and *G. chorda* showed alleviated gastrointestinal dysfunction [38].

### 3.6. Assessment of Inflammatory Factors in G. coronopifolia Synbiotic

Table 2 shows IL-8 level in Caco-2/THP-1 cells after induction with TNF-α (20 ng/mL) and *G. coronopifolia* synbiotic treatments (0.001–0.05 mg/mL). The IL-8 level decreased significantly to 1.13 and 1.10 ng/mL with 0.001 and 0.005 mg/mL GS1, respectively. Simultaneously, the IL-8 level decreased greatly to 0.56 and 1.08 ng/mL with 0.005 and 0.01 mg/mL GS2 treatments, respectively, and IL-8 sharply to 1.11 ng/mL with 0.01 mg/mL GS3 treatment. Thus, *G. coronopifolia* synbiotics efficiently reduced IL-8 production induced by TNF-α, diminishing the inflammatory response, and thereby preventing an intestinal cell inflammatory response. The results agree with treatment with 10 µg/mL methanolic extract of red seaweed, *G. changii* during differentiation of U937 cells, which intensely inhibited the TNF-α response level [39]. Many diseases are due to inflammation. Ulcers in the gastrointestinal tract are typical because of the inflammation in the intestinal tract. This problem affects at least 5% to 7% of the worldwide population [40]. Hence, effective and successive treatment with *G. coronopifolia* synbiotics may be effective for gastrointestinal disorders.

## 4. Conclusions

We still lack healthy food products that combine prebiotics and probiotics. In this study, we used *G. coronopifolia* extract as marine-based seaweed prebiotics and mixed with several probiotics (*Lactobacili* and *Bifidobacteria*) to form natural *G. coronopifolia* synbiotics, which was verified and suitable for maintaining the health of intestinal tract. *G. coronopifolia* synbiotics are nontoxic and effectively inhibited the production of ROS under the oxidative stress of intestinal cells, reduced the damage of cells with oxidative stress, and inhibited the inflammatory response in cells. They also protected and upheld the function of intestinal cells. The development of a combination of marine-based *G. coronopifolia* synbiotics could be used to treat gastrointestinal diseases caused by improper diet or aging of the human body to promote intestinal health. We describe a fundamental, in vitro *G. coronopifolia* synbiotic study that is appropriate for the assessment of intestinal health functions. However, further examinations are required to analyse the several unidentified phytochemical and bioactive compounds of *G. coronopifolia* synbiotics and their biological mechanisms.

## Figures and Tables

**Figure 1 foods-08-00623-f001:**
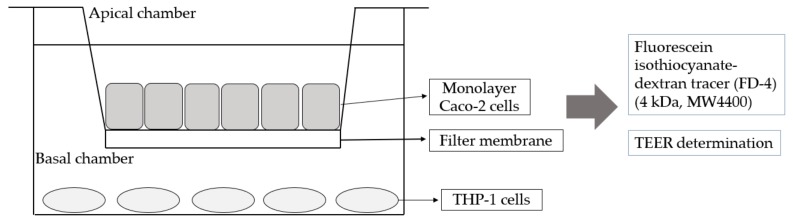
Schematic diagram of a Transwell co-culture system of small intestinal cells (Caco-2) cells and human acute monocytic leukemia (THP-1) cells.

**Figure 2 foods-08-00623-f002:**
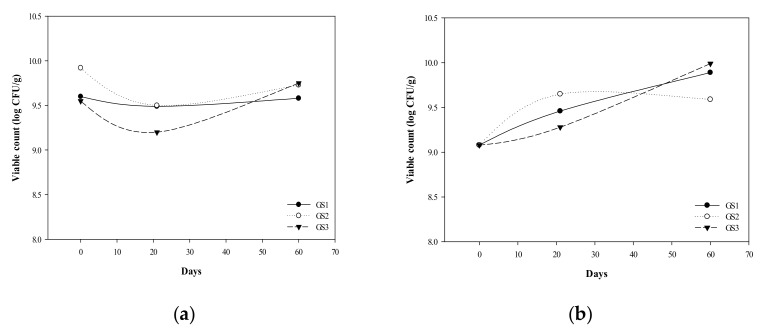
Effect of storage temperature (**a**) 4 °C and (**b**) −20 °C on the stability of *G. coronopifolia* synbiotics.

**Figure 3 foods-08-00623-f003:**
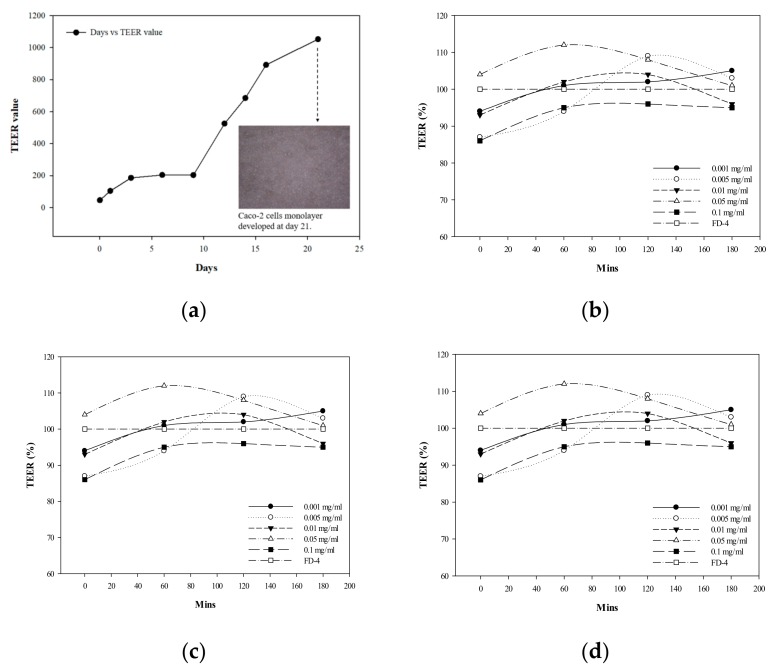
Transepithelial electrical resistance (TEER) values for (**a**) Caco-2 cell monolayer developed at day 21, and Caco-2 cells with the (**b**) *G. coronopifolia* synbiotic (GS1) (**c**) *G. coronopifolia* synbiotic (GS2) and (**d**) *G. coronopifolia* synbiotic (GS3).

**Figure 4 foods-08-00623-f004:**
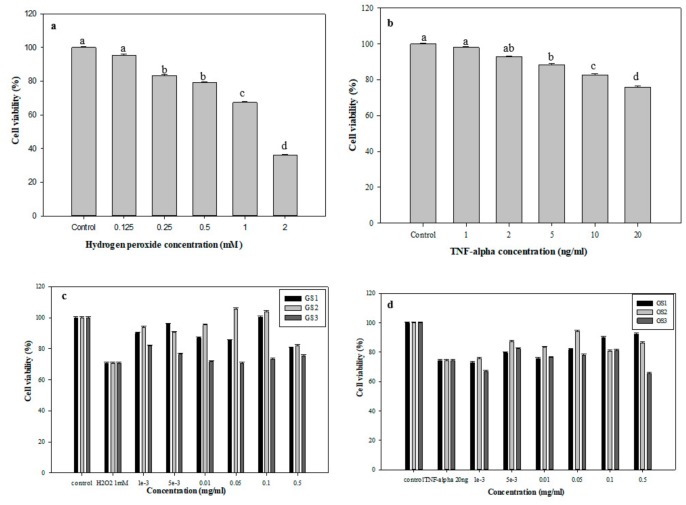
Cell viability (%) of THP-1 cells after induction by different doses of (**a**) H_2_O_2_; (**b**) TNF-α; treatment with *G. coronopifolia* synbiotics and induced by (**c**) H_2_O_2_ and (**d**) TNF-α. Means with the same letters in a column do not differ significantly (*p* < 0.05) (*n* = 3).

**Figure 5 foods-08-00623-f005:**
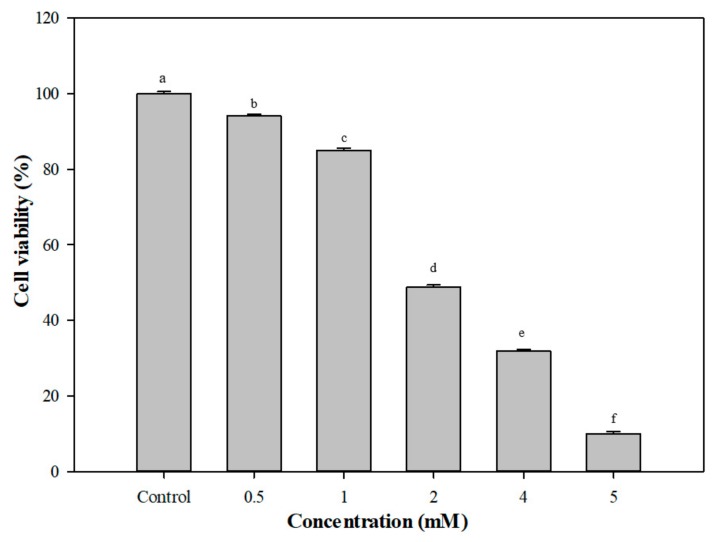
Cell viability of Caco-2 cells after induction with different concentrations of H_2_O_2_. Means with the same letters in a column do not differ significantly (*p* < 0.05) (*n* = 3).

**Table 1 foods-08-00623-t001:** ROS production (fluorescent intensity) of Caco-2 cells after treatment with *G. coronopifolia* synbiotics (mg/mL) and induced by H_2_O_2_ and TNF-α.

Concentration	Control	0.5 mM H_2_O_2_	0.001	0.005	0.01	0.05	Control	20 ng/mL TNF-α	0.001	0.005	0.01	0.05
GS1	100	105.4	84.3	81.1	67.3	63.3	100	124.5	95.2	103.6	93.0	106.6
GS2	100	105.4	62.8	89.5	81.0	48.8	100	124.5	60.6	76.2	81.1	72.8
GS3	100	105.4	74.1	66.0	56.2	68.6	100	124.5	78.2	57.7	63.0	91.3

* GS1 (Strains: dried extract = 30:70); GS2 (Strains: dried extract = 50:50); GS3 (Strains: dried extract (70:30).

**Table 2 foods-08-00623-t002:** Interleukin-8 (IL-8) levels in Caco-2/THP-1 cells after induction by TNF-α and treatment with different concentrations of *G. coronopifolia* synbiotics (mg/mL).

Concentration	Control	TNF-α (ng/mL)	0.001	0.005	0.01	0.05
GS1	0.34	1.47	1.13	1.10	1.77	1.12
GS2	0.34	1.47	1.38	0.56	1.08	1.61
GS3	0.34	1.47	1.62	1.62	1.11	1.50

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
