# Peer review of "Feasibility of Using Seaweed (Gracilaria coronopifolia) Synbiotic as a Bioactive Material for Intestinal Health"

_foods, 2019, doi:10.3390/foods8120623_

Round 1

Reviewer 1 Report

Po-Hsien Li and co-workers present the manuscript entitled “Feasibility of using seaweed (Gracilaria coronopifolia) synbiotic as a bioactive material for intestinal health”, where they explore the effect of G. coronopifolia alone (prebiotic) or mixed with Bifidobacterium and Lactobacillus (synbiotic) in the levels of ROS in Caco-2 cells. The authors found that the treatment with the symbiotic could inhibit the production of ROS and of IL-8.

This is an interesting manuscript, however, there are several issues that need to be carefully revised:

I would suggest a professional English revision once, in most of the sections throughout the manuscript, the text is almost unreadable. There are a lot of sentences that are ambiguous and should be rephrased, for instance:

Lines 33-34: Seaweeds have played an important aquaculture as traditional remedies for many centuries for treating hyperthyroidism and other glandular diseases

Lines 38-39: The species has salt tolerance because the osmolarity of the cytoplasm will adjust automatically

Lines 40-41: “high in minerals”? Do the authors mean “rich in minerals”?

Lines 41-43: The natural products, metabolites, isolated from marine seaweeds are important sources of bioactive compounds that could be developed into treatments for some diseases.

And so on.

Lines 72-74: “In recent years, studies have shown that when the body is exposed to extensive oxidative stress, it generates free radicals and reactive oxygen species (ROS), which have a great impact on human health [17]”. This has been known for decades, not only in recent years. Please rephrase. Line 118: “were trypsinized by aspirating the supernatant”. Please rewrite the sentence. 2.3. Where the different probiotic strains added in the same ratio? Please clarify. 2.4. Cell culture: Why did the authors add FBS to neutralize trypsin, when culture medium in enough? Did the authors cultivate cells without adding any antimicrobial to the culture medium?  Lines 133-134: “THP-1 cells, which were stimulated with lipopolysaccharides (LPS) and tumour necrosis factor α (TNF-α) for 24 hr”. Please add the concentrations of TNF-α and LPS. 2.6. Lines 143-144: “…and 100 μL of different concentrations (0.1-1000 μg/mL) of sample solutions was added…” How did the authors prepare the solutions. This is not clear in the methods section. Please clarify.

Line 146: Did the authors add DMSO without previously removing the culture supernatant? Please clarify

If the MTT assay measures the activity of cell dehydrogenases, why a blank with no cells?

The formula for cell viability should be: (Abs sample X 100)/Abs Control

Please rewrite the whole section. The methodology is not clear and very confuse.

 Line 213: DO the authors mean “remained 106 cfu/g”?

The number of bacteria increased with a lower storage temperature?

 The procedure regarding cells co-culture is not very clear. I suggest the authors to include of a scheme of the procedure, which would definitely enrich the manuscript.  Figure 3: Please add the statistical analysis to the graphs.  Please remove Table 2 from the conclusions section, and put it within Results and Discussion section. How many independent assay have been performed?  The authors should further explore their results, and not only limit to say that G.  coronopifolia synbiotics may be effective for gastrointestinal disorders. Additionally, the results and discussion section is mostly limited to the description of the results obtained by the authors, and the comparisons with previous works are few or inexistent. The discussion should be enriched in order to valorize the work and enrich the manuscript.

Author Response

Response to Reviewer 1 Comments

We appreciate the reviewer comments very much and have revised the manuscript accordingly. The specific changes we have made in the revised manuscript are, as follows:

Point 1: I would suggest a professional English revision once, in most of the sections throughout the manuscript, the text is almost unreadable. There are a lot of sentences that are ambiguous and should be rephrased.

Response 1:

Thanks for the comment. Some grammatical errors, verb tense, singular/plural, and improper words had been modified and revised. Moreover, this paper has been edited by a native English speaker prior to submission.

Point 2: Lines 72-74: “In recent years, studies have shown that when the body is exposed to extensive oxidative stress, it generates free radicals and reactive oxygen species (ROS), which have a great impact on human health [17]”. This has been known for decades, not only in recent years. Please rephrase. Line 118: “were trypsinized by aspirating the supernatant”. Please rewrite the sentence.

Response 2:

Thanks for the comment and positive evaluation. The sentence has been re-written and redraft. All changes are marked in yellow background in the revised manuscript.

Point 3: 2.3. Where the different probiotic strains added in the same ratio? Please clarify.

Response 3:

Thanks for the comment. The five probiotic strains were with the ratio of 20:20:20:20:20, corresponding to 1.0 × 108 CFU/mL.

Point 4: 2.4. Cell culture: Why did the authors add FBS to neutralize trypsin, when culture medium in enough? Did the authors cultivate cells without adding any antimicrobial to the culture medium? 

Response 4:

Thanks for the comment. Trypsin digests the protein molecules (matrix/ cementing material) which makes cells to stay adherent on solid surface. FBS contains protease inhibitors principally α1-antitrypsin, which will inhibit the trypsin activity. Before the addition of trypsin, cells should be washed with PBS to remove any leftover FBS, because this could hinder the trypsinisation process. The cells cultivated in Dulbecco's modified eagle's medium (DMEM) (glutamine, high glucose, Sigma) containing 10% fetal bovine serum (FBS) and supplemented with 1% of antimicrobial (Antibiotic-Antimycotic, 100X, 15240062, Gibco U.S.A.) at 37 °C with 5% CO2. All changes are marked in yellow background in the revised manuscript.

Point 5: Lines 133-134: “THP-1 cells, which were stimulated with lipopolysaccharides (LPS) and tumour necrosis factor α (TNF-α) for 24 hr”. Please add the concentrations of TNF-α and LPS.

Response 5:

Thanks for the comment. The concentrations of TNF-α and LPS had been added. All changes are marked in yellow background in the revised manuscript.

Point 6: 2.6. Lines 143-144: “…and 100 μL of different concentrations (0.1-1000 μg/mL) of sample solutions was added…” How did the authors prepare the solutions? This is not clear in the methods section. Please clarify.

Response 6:

Thanks for the comment. A stock solution was prepared, and next, the stock solution was diluted by using the dilution factor to prepare for the sample solutions with different concentrations.

Point 7: Line 146: Did the authors add DMSO without previously removing the culture supernatant? Please clarify.

Response 7:

Thanks for the comment. DMSO was added into each well on 96-well plates without previously removing the culture supernatant to terminate the reaction before the absorbance was read by an ELISA reader.

Point 8: If the MTT assay measures the activity of cell dehydrogenases, why a blank with no cells?

Response 8:

Thanks for the comment. The sample was replaced with the culture medium as the control group, while the blank group was the culture medium containing no cells and samples to further deducted of the blank value during calculations.

Point 9: The formula for cell viability should be: (Abs sample X 100)/Abs Control

Response 9:

Thanks for the comment. The formula has been re-written and redraft. All changes are marked in yellow background in the revised manuscript.

Point 10: Please rewrite the whole section. The methodology is not clear and very confuse.

Response 10:

Thanks for the comment. The ‘Materials and Methods’ section has been re-written and redraft. Some grammatical errors, verb tense, singular/plural, and improper words had been modified and revised.

Point 11: Line 213: DO the authors mean “remained 106 cfu/g”? The number of bacteria increased with a lower storage temperature?

Response 11:

Thanks for the comment. The number of bacteria of probiotic products containing live bacteria needs to be ≥106 cfu/g to exert their activity; therefore 106 cfu/g is used as a screening condition for storage. After storage at -20 °C for 60 days, the number of bacteria was still maintained at 106-108 cfu/g for GS3 and GS1 synbiotics.

Point 12: The procedure regarding cells co-culture is not very clear. I suggest the authors to include of a scheme of the procedure, which would definitely enrich the manuscript.

Response 12:

Thanks for the comment. The schematic diagram of a Transwell co-culture system of Caco-2 cells and THP-1 cells (Figure 1) had been added into section 2.5 ‘Co-culture of Caco-2 and THP-1 cells’.

Point 13: Figure 3: Please add the statistical analysis to the graphs.

Response 13:

Thanks for the comment. Figure 3 had been modified and added with different superscript letters for the significant differences.

Point 14: Please remove Table 2 from the conclusions section and put it within Results and Discussion section.

Response 14:

Thanks for the comment. Table 2 had put it within ‘Results and Discussion’ section.

Point 15: How many independent assays have been performed?

Response 15:

Thanks for the comment. Each sample was carried out with three replications.

Point 16: The authors should further explore their results, and not only limit to say that G. coronopifolia synbiotics may be effective for gastrointestinal disorders. Additionally, the results and discussion section are mostly limited to the description of the results obtained by the authors, and the comparisons with previous works are few or inexistent. The discussion should be enriched in order to valorize the work and enrich the manuscript.

Response 16:

Thanks for the comment. The potential mechanisms of probiotics action in intestinal tract are, for instance, modification of intestinal microbiota, improvement of colonic physicochemical conditions, production of anticancerogenic and antioxidant metabolites against carcinogenesis, a decrease in intestinal inflammation, and the production of harmful enzymes. Lactobacillus spp. and Bifidobacterium spp. are two of the lactic acid bacteria (LAB) that belong to the natural intestinal microbiota. Reduction of these bacteria may contribute to a low degree of inflammation in which the level of proinflammatory cytokines (Moraes-Filho et al., 2015). Hence, maintaining the microbiota stability in the intestinal tract is crucial. The ‘Materials and Methods’ section has been re-written and redraft. All changes are marked in yellow background in the revised manuscript.

References

Moraes-Filho, J.P., Quigley, E.M. the Intestinal Microbiota and the Role of Probiotics in Irritable Bowel Syndrome: A review. Arq. Gastroenterol. 2015, 52, 331–338.

Reviewer 2 Report

Authors Li et al, developed a sea-weed based synbiotics and verified the efficacy on intestinal epithelial cells that promote health on intestinal health.

<Abstract>

Line 13-14: This sentence is complex and can be simplified into two sentences.

<Methods>

Purchase/Source of probiotics is missing.

Quantiy/CFU utilized for each probiotics needs to be included instead of 20% ratio which may affect the results.

Reason to choose these specific probiotics

Probiotic strain numbers are missing.

<Results>

Why didn't authors assess the individual probiotic/s or only seaweed effects on the intestinal epithelium.

<Overall>

Genus names should be italic

<References>

More recent references are needed.

Author Response

Response to Reviewer 2 Comments

We appreciate the reviewer comments very much and have revised the manuscript accordingly. The specific changes we have made in the revised manuscript are, as follows:

Point 1: <Abstract> Line 13-14: This sentence is complex and can be simplified into two sentences.

Response 1:

Thanks for the comment and positive evaluation. The sentence has been re-written and redraft. All changes are marked in yellow background in the revised manuscript.

Point 2: <Methods> Purchase/Source of probiotics is missing.

Response 2:

Thanks for the comment. The probiotic strains were purchased from the Bioresource Collection and Research Centre of the Hsinchu Food Industry Research and Development Institute. The sentence has been re-written and redraft. All changes are marked in yellow background in the revised manuscript.

Point 3: Quantity/CFU utilized for each probiotic needs to be included instead of 20% ratio which may affect the results.

Response 3:

Thanks for the comment. Quantity/CFU for the probiotic was maintained at absorbance wavelength at 610 nm reached 0.8 (1.5 × 108 CFU/mL), suitable for further analysis. The sentence has been re-written and redraft. All changes are marked in yellow background in the revised manuscript.

Point 4: Reason to choose these specific probiotics.

Response 4:

Thanks for the comment. The potential mechanisms of probiotics action in intestinal tract are, for instance, modification of intestinal microbiota, improvement of colonic physicochemical conditions, production of anticancerogenic and antioxidant metabolites against carcinogenesis, a decrease in intestinal inflammation, and the production of harmful enzymes. Lactobacillus spp. and Bifidobacterium spp. are two of the lactic acid bacteria (LAB) that belong to the natural intestinal microbiota. Maintaining the microbiota stability in the intestinal tract is crucial. Furthermore, Lactobacillus and Bifidobacterium show an anti-inflammatory effect in the intestine. Therefore, these two specific probiotics were choosed as the probiotics strains in this study. 

Point 5: Probiotic strain numbers are missing.

Response 5:

Thanks for the comment. The probiotic strain number was added into the revised manuscript, which is the Bifidobacterium bifidums (BCRC 11844), Bifidobacterium longum subsp. infantis (BCRC 14602), Bifidobacterium longum subsp. longum (BCRC 11847), and Lactobacillus acidophilus (BCRC 10695), Lactobacillus delbrueckii subsp. bulgaricus (BCRC 16053). The sentence has been re-written and redraft. All changes are marked in yellow background in the revised manuscript.

Point 6: <Results> Why didn't authors assess the individual probiotic/s or only seaweed effects on the intestinal epithelium.

Response 6:

Thanks for the comment and positive evaluation. Previous studied had studied the individual probiotic/s or only seaweed effects towards the intestinal epithelium (Lin et al., 2019; Park et al., 2002; Quinn et al., 2018). However, there is no study mentioned about the effect of Gracilaria coronopifolia with Lactobacillus spp. and Bifidobacterium spp. synbiotic on the intestinal tract. Hence, our study was focused on the intestinal health using seaweeds-based Gracilaria coronopifolia synbiotic by Caco-2 cells model.

References

Lin, T., Liu, X., Xiao, D., Zhang, D., Cai, Y., Zhu, X. Lactobacillus spp. as probiotics for prevention and treatment of enteritis in the lined seahorse (Hippocampus erectus) juveniles. Aquaculture. 2019. 503, 16-25.

Park, J. H., Um, J. I., Lee, B. J., Goh, J. S., Park, S. Y., Kim, W. S., Kim, P. H. Encapsulated Bifidobacterium bifidum potentiates intestinal IgA production. Cellular Immunology. 2002. 219(1), 22-27.

Quinn, E. M., Slattery, H., Thompson, A. P., Kilcoyne, M., Joshi, L., Hickey, R. M. Mining Milk for Factors which Increase the Adherence of Bifidobacterium longum subsp. infantis to Intestinal Cells. Foods. 2018. 7(12).

Point 7: <Overall> Genus names should be italic.

Response 7:

Thanks for the comment and positive evaluation. Genus names in the manuscript had been italic. We had been modified the entire manuscript according to the reviewer’s comments. All changes are marked in yellow background in the revised manuscript.

Point 8: <References> More recent references are needed.

Response 8:

Thanks for the comment. A few recent references had been added in the manuscript. All changes are marked in yellow background in the revised manuscript.
